# Impact of Sports Mass Media on the Behavior and Health of Society. A Systematic Review

**DOI:** 10.3390/ijerph16030486

**Published:** 2019-02-08

**Authors:** Puertas-Molero Pilar, Marfil-Carmona Rafael, Zurita-Ortega Félix, González-Valero Gabriel

**Affiliations:** Departament of Didactics of Musical, Plastic and Corporal Expressión, University of Granada, 18071 Granada, Spain; pilarpuertas@correo.ugr.es (P.-M.P.); rmarfil@ugr.es (M.-C.R.); felixzo@ugr.es (Z.-O.F.)

**Keywords:** sports journalism, media effects, healthy lifestyle, social influence, behavioral changes

## Abstract

The presence of sport in the media has grown exponentially over the last few decades. As a result, the influence of the media on the concept of physical activity within society and the collective and individual values it purports is indisputable. The mass media tends to follow a specific pattern when representing sport, this includes broadcasting of sport competitions and presentation of elite athletes as contemporary legends. A broad range of teaching and research opportunities are available in the field of media education. For this reason, we conducted a systematic review of international studies (Web of Science and Scopus) published between 2007 and 2018, focusing on the effects and influence of sports content on the audience. The Preferred Reporting Items for Systematic Reviews and Meta-Analyses (PRISMA) statement provided a framework for the analysis of included papers. The study incorporated an initial sample of 313 research articles that discussed the importance of emotional factors with regards to perceptual processes. Furthermore, links with various behavioral indicators were identified, such as competitiveness, violence, self-improvement linked to effort, stereotypes of beauty and health care.

## 1. Introduction

The media conditions society’s vision and understanding of sport. In a certain way, the media provides a narrative that exerts a didactic influence on the concept of sport and physical education [1]. A conception of the mass media is provided by Hyuk-Lee et al. (2009) [2] as sources of information about a recent event, which generate a growing interest. Furthermore, the transmitted message is developed according to criteria of topicality, objectivity and simplicity, which aim to produce a message with the widest reach possible [3,4]. In this sense, what is determined as “newsworthy” content in sport is therefore governed by these criteria.

Through various types of media, a high level of interest of spectators in sport in general is generated and research makes it possible to identify the conditions in which the modern culture and communication industry develops [5,6]. The contemporary digital culture presents a complex network of “hyper mediations” [7]. In addition, the didactic potential and influence of the content of sports information is relatively unexplored territory.

The rise of different social sport networks in current society enables journalists from different media groups to publish information assiduously. In this way, they can directly contact consumers and exert a greater influence [8,9]. Despite being an attractive medium of information, used by journalists to express, disseminate and contrast their own opinion with those from other perspectives, reporters are limited in the content they can diffuse [10,11,12].

The availability of different means of communication has the advantage of allowing various channels to be used simultaneously to share information. An important aspect of the digital culture is that it attracts an extreme level of interest in media content which generates undoubted influence on a large number of different areas, this is known as the “hostile media effect” [13,14,15]. This has been previously examined with regards to the influence of viewing television media by examining specifically the gaze of children [16,17,18]. The contemporary approach to this line of work instead seeks to examine strategies for media and digital literacy [19]. The importance of this is demonstrated when the radical increase of journalists in the field of sports information is considered, with journalists now seeking to acquire greater personal prominence through live broadcasting, through which many sports are now reproduced daily on television sets [20,21,22].

### 1.1. Sports Information and Emotions

Given the new demands in the digital age, match calendars are now adapted bearing in mind the optimal dates and time to reach the largest audience and ensuring that main events from different sports do not overlap in order to meet demands [8,23]. This has led to changes at the cultural level, culminating in a magnification of the repercussions experienced by athletes in the face of both victory and defeat. These types of events draw in the audience and enables them to experience the feeling of being members of a group and to identify with ‘the colors of flag’ [24,25,26].

### 1.2. Regulation

From the point of view of the content of sports information, another aspect in which it can exert influence is on the regulation of sports and in the decision making of referees (break taking in tennis). Furthermore, media content can be indirectly persuasive, for instance, through the use of the iconography found in different stadiums, which includes a high number of advertising posters in addition to the continuous provision of attention to all the details and animations likely to satisfy spectators [27,28].

Persuasion not only describes the attraction to consume various sporting events but is also a force that influences society at both a behavioral and sentimental level. As stated by Francisco et al. (2013) [29] and Stirling et al. (2012) [30], viewers demand that the spectacle fulfills certain parameters of physical appearance. This could cause some athletes to develop eating disorders, particularly in sports such as gymnastics or athletics, where athletes feel pressured to continue lowering their weight in response to cultural standards set by the society in which they perform [31,32].

However, publicity of athletes also has a positive influence on spectators. Through multiple social networks athletes present an image linked to positive values, which motivates citizens to make positive behavioral changes in their lives such as engaging in more physical exercise or consuming a balanced diet. This can lead to healthier habits being acquired by members of the population [33,34].

Both the athletes who are being observed and the spectators who are consuming information live in a society immersed in new technologies. This has awakened the interest of many researchers in understanding the drivers of behavioral change in the age of mass media. For this reason, the scope of the present systematic review aims to improve understanding of the scientific evolution of this subject, including examining the healthy or harmful effects that the sports media can exert within society.

## 2. Materials and Methods

The Preferred Reporting Items for Systematic Reviews and Meta-Analyses (PRISMA) statement for systematic reviews was used to structure the present review and increase its integrity [35,36]. Studies were classified and coded by the authors through independent evaluation. Studies were deleted when independent codings failed to coincide on a single occasion. Reliability of the coding was checked by dividing the total number of matches by the total number of categories proposed for the study and multiplying this outcome by 100. Degree of agreement was required to exceed 80% for a study to be included.

### Process

He systematic review was carried out during the months of December and January 2007–2018 and focused on studies which analyzed the influence of the sports media on spectators. For this purpose, Web of Science (WOS) and SCOPUS were employed as the main search engines. The time range was delimited as 2007–2018 and the keywords “Media”, “Sport” and “Influence” were searched with the Boolean operator “and” being used.

All research published in either the English or Spanish languages were considered, resulting in a total of 313 studies being obtained for further examination. In order to specify the theme of the work, the research domains “Sport Sciences”, “Communication” and “Psychology” were targeted, with other less relevant areas to the study objective being disregarded. After refining the literature search, the final study sample was selected according to the following inclusion criteria: (i) studies in which at least one means of communication was used to spread information; (ii) designed to address the types of influences exerted by the media; (iii) included media designed to address the public; and (iv) used a cross-sectional or experimental research design.

In processing the data, the title and summary of the sample were first critically read in order to confirm that they met the inclusion criteria. Following this, the full text was critically read to confirm that the article met the objectives of the present study. This resulted in 277 articles being excluded due to a lack of concordance in the coding of independent evaluators, or the studies not meeting the established conceptual and methodological criteria. This left a final sample of 36 scientific articles which constituted the basis of the present study, as shown in Figure 1.

## 3. Results

### 3.1. Data from Studies Selected for Systematic Review

With regards to the extraction and codification of data, the following details were recorded: (1) authors and year of publication; (2) population; (3) type of sport analyzed; (4) media used; (5) instruments used to measure the influence exerted, and (6) type of research developed.

The data presented in Table 1 shows that the majority of the included studies (75%) used a cross-sectional methodological design. Research on this topic has been conducted in 12 different countries, with the USA being the country in which most relevant work has been carried out (*n* = 15), followed by Australia which was home to three included studies. Football (34.54%) was the sport which has received the most interest. In addition, television (56%) was the medium to have been most frequently analyzed by research studies.

In relation to the type of persuasion exercised by the media, a number of categories were established, which encompassed the persuasive approach taken, with studies being removed from this analysis when they did not share any common approaches with other studies. In Table 2, the different types of influences can be observed, alongside the frequency and proportion with which they appear in the selected publications. The types of influence are differentiated according to the categories health, emotional, physical and violence. Health influences are understood as those that contribute to the development of active lifestyles and well-being. Emotional influences describe responses that arouse nationalistic feelings and the sensation of belonging to a group. Physical influences are those that impact upon body image. Finally, influences relating to violence describe the development of disruptive attitudes and consumption effects, such as those that incite society to acquire products such as trainers.

Table 2 shows that within the selected studies, emotional influences (33.3%) were the most commonly studied, followed by the categories of health and consumption (22.2%), with violence being the type of influence to receive the least attention in research studies (2.8%).

### 3.2. Data from Studies Selected for Systematic Review

In the last decade, 313 articles have been indexed in WOS and SCOPUS, examining the influence of sports media on society. Of these, 36 were selected for inclusion in the present study following application of the inclusion and exclusion criteria. This describes 11.05% of overall production. Figure 2 outlines the comparison between the production of articles per year and those which were selected from each examined database (WOS and SCOPUS), to be included in the present analysis.

The production of articles dealing with the topic analyzed has ascended over the last decade, reaching its highest point in 2017 (*n* = 58) when most articles were published, followed by 2016 (*n* = 43). The greatest decline in publication of relevant articles is detected in 2008 (*n* = 9), though 2014 (*n* = 23) should be acknowledged as a year in which interest in the topic decreased as publications relating to the topic had been high prior to this year.

## 4. Discussion

The emergence of the sports media over recent years has made it a topic of interest for research. They have experienced a huge expansion, which has facilitated their reach to the majority of the population. Through this expansion they exert multiple influences, which is reflected in the behavior of society [3,25,30,32,61].

A study conducted by Atwell-Seate et al. (2016) [38] examined the prominent emotional and violent influences during the aftermath of sporting events, finding that emotional factors play an important role on the feelings of spectators. In this way, they are able to participate in the successes and failures of their teams, generating a feeling of shared social identity, which can lead to nationalistic feelings [24,26,43,47,51]. In the same sense, results of investigations reported by Cummins et al. (2017) [27] and Kim et al. (2011) [23], reveal that the more intense the feelings experienced in sports stadiums, the greater viewers immerse themselves in the event. This demonstrates the great potential influence that the producers responsible for these events can have. However, the experience of such exaggerated feelings can lead to violent and aggressive attitudes towards the opponents of the other teams [40]. These feelings influence the sociability of spectators. From a negative perspective, such excessive emotional reactions can lead to fanaticism, which can lead to violence. This shows that sports media can have both positive and negative influences on society and values.

In the same way, studies reported by Kennard et al. (2018) [50] and Ličen et al. (2008) [52], identified that the subjective opinion of commentators directly influenced the behavior of spectators, specifically, increasing the sympathy or rejection directed towards a certain team by increasing feelings of euphoria before victories and defeats. This shows that the sports media influence the behavior and personality of society, particularly amongst younger individuals who are more vulnerable.

With regards to the influence exerted by the mass media on the acquisition of healthy habits, research carried out by Gao (2012) [46] demonstrated that favorable attitudes towards engaging in exercise can be generated in university students by exposing them to sports press images. Furthermore, these attitudes were maintained over time.

Similar findings have been reported following studies conducted by Gietzen et al. (2017) [34] and Li et al. (2015) [53]. These studies confirmed that viewers were motivated to improve their habits after watching famous entities perform physical exercise and consume a balanced diet [55]. Advertising attracts society’s attention and stimulates the need for physical activity by strongly influencing our habits and lifestyles.

Sports mass media have the ability to promote physical activity by influencing social relations [57,62]. Their great expansion has made them available to the entire population. In this sense, Brown et al. (2010) [33] confirmed that exhibiting the private life of athletes helps maintain positive feelings towards engagement in physical activity and sports, and encourages the rejection of drugs [37]. This feeling of empathy is a key factor in the promotion of healthy habits from an early age.

However, when considering the influences exerted by the media on physical norms, it is necessary to highlight that continuous exposure to the bodies of athletes through the media has produced a distorted perception of beauty, which causes body image dissatisfaction within the general population [39,49]. In research developed by Nerimi (2015) [54], results revealed that adolescent athletes were less satisfied with their physique than females who reported sedentary habits. This outcome was caused by the pressure placed on these young athletes by society. This pressure instilled in them a desire to reach extreme thinness, which they achieved by engaging in insane and harmful amounts of exercise rather than healthy exercise [45]. The excessive and improper use of sports media generates pressure on athletes to achieve a distorted standard of beauty, which can lead to mental problems (stress, anxiety) or eating disorders.

Cranmer et al. (2017) [42] showed a further influence exercised by various media types on the physical aspect. They highlighted that the media’s portrayal of certain athletes as more closely achieving a certain beauty standard led to them being more popular with and supported by spectators. This media pressure is stronger within the female sports sphere and can produce very high levels of frustration [48,58]. This attitude generated by the media in society has negative repercussions as it leads to feelings of discrimination towards those who are less favored.

With regards to influences on consumption, the media use sportspeople to sponsor their products and influence the desire to acquire these products [59]. To this end, Voorveld et al. (2015) [60] stated that the morning hours are primarily used to promote products as there is a large number of spectators at this hour and strategies such as scarcity can be used [41]. It should be added that the media often uses advertising for personal gain and to increase the sales of their products with little consideration of the impact on people’s lifestyles.

The media do not only exert a significant influence on the acquisition of products. In addition, through new social networks the media facilitates and enables viewers to participate in their own news, whilst also consuming multiple sporting events directly [44,56]. This causes an increase in the number of hours in which viewers consume sports media and increases their identification with their team [8]. Through the above discussion, the results of the present research offer a diverse perspective in relation to the influence of sports information.

## 5. Conclusions

The aim and scope of the present systematic review was to better understand the evolution experienced by the sports media in the last decade, in addition to the influences that these media exert on society and the way in which sport is perceived.

It is important to highlight that many countries have demonstrated interest in examining the research topic presented here, especially in the USA where the majority of previous research has been carried out. However, examination of the two databases included in the present research failed to identify any previous study conducted in Spain. This makes it necessary to highlight the need for future research on the subject to be conducted in the country of the authors of the present study, in order to understand the different influences that the media exert within this population.

When considering the different sport modalities, football continues to be the most frequently researched and followed by spectators, although other sports are shown to be somewhat popular. The majority of research is focused on analyzing the sport of football.

With regards to the promotion of healthy habits through sports media, the present findings highlight positive influences on motivation to make lifestyles changes. However, it is also observed that the majority of strategies for positive behavioral change are accompanied by a different type of persuasion that has becomes a priority in the socioeconomic context. This includes the promotion of products for consumption by spectators.

It is recommended that future research include a more exhaustive analysis of the influence of sports media on the health of spectators. This should include an examination of how the media contribute to changes in lifestyles and how they facilitate the acquisition of these habits, in addition to examining the connection between sports information and a communicative education. This is highly relevant given that mass media are consumed from very early ages and have yet to be comprehensively explored.

Finally, the enormous influence of the treatment of sport in both the traditional and digital media is reflected by the effects produced by various types of media content. The need to promote a critical vision within citizens is highlighted. This should promote a greater tendency towards the perception of positive individual and social health aspects over the creation of inconsistent iconologies or product consumption. Sport in the media follows a model of representation and information treatment, which also impacts upon the values of society.

## Figures and Tables

**Figure 1 ijerph-16-00486-f001:**
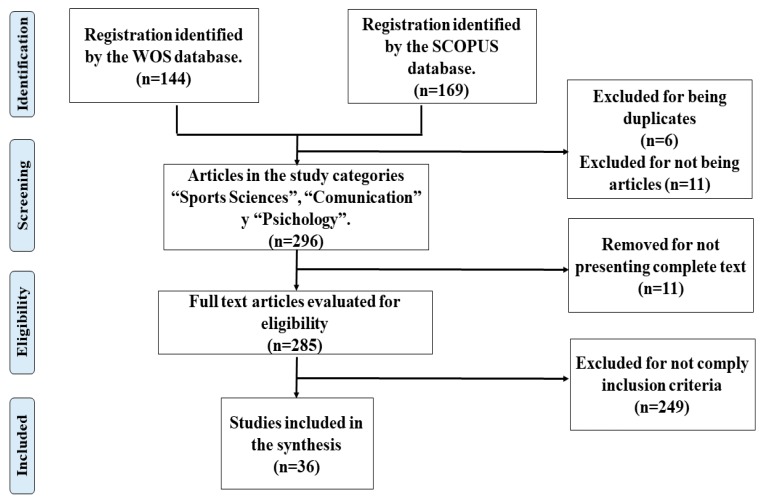
Flowchart of the systematic selection of articles for inclusion in the present study.

**Figure 2 ijerph-16-00486-f002:**
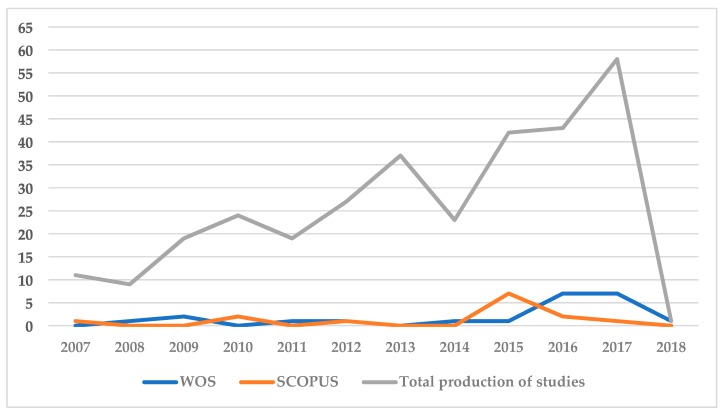
Evolution of scientific production. Vertical axis, number of articles published; Horizontal axis, year of publication; Grey line, total number of relevant articles identified from Web of Science (WOS) and SCOPUS to be analyzed; Blue line, selected articles from WOS; Orange line, selected articles from SCOPUS.

**Table 1 ijerph-16-00486-t001:** Codification of the scientific studies that form the basis of the study.

Authors (year)	Country	Type of Research	Sample	Population (Rank 7 Average Age)	Sport	Type of Media	Instruments *	Type of Influence
Albalawi et al. (2015) [37]	Saudi Arabia	Cross-sectional	14	Athletes (-)	Various sports	Twitter	Classification lists	Health
Atwell-Seate et al. (2016) [38]	USA	Experimental	182	University students (19.26 ± 1.75)	Football	Television	MUP MCP MSM	Emotional
Bissell et al. (2010) [39]	USA	Cross-sectional	117	Adolescents (7–16)	Gymnastics Volleyball Basketball Tennis Horse racing	Television press release	Structured survey	Physical
Birkner et al. (2016) [24]	Germany	Cross-sectional	14	Atlhletes (-)	Football	Television press release	Inquiry	Emotional
Brady et al. (2007) [40]	USA	Cross-sectional	319	University students (18–20)	Contact sports	Television	ATVS ad-hoc	Violence
Brown and De Matvink (2010) [33]	Argentina	Cross-sectional	360	University students (-)	Football	Television	Structured survey	Health
Burk et al. (2016) [33]	Germany	Cross-sectional	8654	General population (16–60)	Football	Social media Television press release	Ad-hoc	Health
Checchinato et al. (2015) [41]	Italy	Cross-sectional	375	Athletes (-)	Football	Internet	Ad-hoc	Consumption
Cranmer et al. (2017) [42]	USA	Experimental	49	University students (19.18 ± 1.78)	Football	Television	FKGLT	Physical
Cummnis et al. (2017) [27]	USA	Experimental	122	University students (20.91 ± 1.93)	Football	Radio	MTM	Emotional
Devlin et al. (2016) [43]	USA	Experimental	979	General population (16–65)	Football	Television	Ad-hoc	Emotional
Devlin et al. (2017) [8]	USA	Cross-sectional	715	General population (18–65)	Various sports	Television	Personality inventory SSIS Ad-hoc	Consumption
English (2016) [44]	Australia	Cross-sectional	364103	Sports journalist (-) Press articles (-)	Football Rugby Hockey	Twitter	Structured survey Ad-hoc	Consumption
Fortest et al. (2015) [45]	Brasil	Cross-sectional	133	Athletes (12–17)	Gymnastics Athletics	Television press release	SATAQ-3 BSQ	Physical
Gao (2012) [46]	China	Cross-sectional	398	University students (-)	Football Basketball	Television Internet	SQIICPSISB	Health
Gietzen et al. (2017) [34]	USA	Cross-sectional	2793	Adolescents (14.4)	Footing Football	Television	Ad-hoc	Health
Gong (2017) [47]	USA	Experimetal	12	Fanatics (20–25)	Football	Television	Interview	Emotional
Godoy-Pressland (2016) [48]	England	Cross-sectional	22,717	Athletes (-)	Football	Press release	Content coding	Physical
Hardy (2015) [49]	Canada	Cross-sectional	9	Athletes (-)	Rugby	Television	Structured survey	Physical
Kennard et al. (2018) [50]	Australia	Cross-sectional	52	Athletes (-)	Rugby	Television	Content coding	Emotional
Kim et al. (2011) [23]	Korea	Cross-sectional	1444	General population (18–67)	Football	Television	EHESV	Emotional
Knobloch et al. (2009) [51]	USA	Cross-sectional	113	University students (-)	Football	Television	SSIS PANAS	Emotional
Knoll et al. (2014) [26]	Germany	Experimental	328	General population (29.03 ± 12.48)	Football	Television	Ad-hoc	Emotional
Licèn et al. (2008) [52]	Slovenia	Cross-sectional	2	Sports Commentators	Basketball	Television	Classification of content	Emotional
Li et al. (2015) [53]	China	Cross-sectional	16	General population (24–42)	Football Swimming Basketball	Television	Semi-structured survey	Health
Mastro et al. (2012) [20]	USA	Cross-sectional	244	University students (22–26)	Hockey Gymnastics Football Basketball	Press	Ad-Hoc	Consumption
Mudrick et al. (2017) [10]	USA	Experimental	544	Adults (35.3 ± 11.2)	Basketball	Television	MRNI-R	Emotional
Mudrick et al. (2016) [32]	USA	Experimental	630	University students (18–33)	Football	Twitter Facebook	Ad-hoc	Emotional
Nerini (2015) [54]	Italy	Cross-sectional	67 68	Athletes (10–14) Sedentary individuals (11–14)	Ballet	Television Press	SATAQ-3 BSQ	Health
Ren (2017) [55]	China	Cross-sectional	172	Adolescents (-)	Various sports	Television press release APP	Ad-Hoc	Health
Rubenking (2016) [56]	USA	Cross-sectional	570	University students (19.42 ± 2.28)	Various sports	Television	ITQ MPI	Consumption
Sherwood et al. (2017) [4]	Australia	Cross-sectional	37	Sports organizers (20–59)	Tenis Cricket Rugby	Television	SGB NVivo	Consumption
Smith et al. (2015) [57]	Australia	Cross-sectional	46	General population (18–65)	Various sports	Television	Semi-structured survey	Health
Swami et al. (2009) [58]	England	Experimental	81 44	Athletes (24–29) Sedentary individuals (26–28)	Athletics Martial	Television	SATAQ-3 PFRS ISAS	Physical
Thompson et al. (2016) [59]	Australia	Cross-sectional	105	General population (18–54)	Tennis	Facebook Twitter	Ad-hoc	Consumption
Voorveld et al. (2015) [60]	USA	Cross-sectional	273	General population (19–50)	Various sports	Television	VCM	Consumption

Note: *, Measuring instruments used in studies; MUP, Measured uncritical patriotism; MCP, measured critical patriotism; MSM, measured support of militarism; ATVS, attitudes toward interpersonal violence scale; FKGLT, Flesch-Kincaid grade level test; MTM, moment to moment; SATAQ-3, sociocultural attitudes toward appearance questionnaire-3; BSQ, body shape questionnaire; SQIICPSISB, survey questionnaire about influences of image communication paradigm of sports information on sports behaviors of university students; EHESV, experience of hedonic, eudaimonic, and social values; SSIS, sports spectator identification scale; PANAS, positive and negative affect scale; MRNI-R, male role norms inventory–Revised; BSQ, body shape questionnaire; ITQ, immersive tendency questionnaire; MPI, multitasking preference inventory; SGB, professional sport and sport governing body—information subsidies; PFRS, photographic figure rating scale; ISAS, involvement in sporting activity scale; VCM, video consumer mapping.

**Table 2 ijerph-16-00486-t002:** Emergence of different types of influence by the media.

Type of Influences	Frequency of Inclusion in Research Studies	Percentage
Influence on health	*n* = 8	22.2%
Emotional influence	*n* = 12	33.3%
Physical influence	*n* = 7	19.5%
Influence on violence	*n* = 1	2.8%
Influence on consumption	*n* = 8	22.2%

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
