# Peer review of "Impact of Sports Mass Media on the Behavior and Health of Society. A Systematic Review"

_ijerph, 2019, doi:10.3390/ijerph16030486_

Round 1

Reviewer 1 Report

The manuscript focus an intersting topic and includes a table of contents of systematically selected papers. This is said to be a systematic review. For me it seems to be a scoping review, as it does include what the research is about but the synthesis of what can be concluded from the studies is not well developed. 

Major comments:

The paper is a scoping review. No information is given on the strenghts of the conclusions. Some studies were rather small. No meta-analysis was done probably because the diversity of designs, data collected and samples.

The aims, on linme 75-78, are very ambitious. The formulation impliues that quantitative/statistical analysis should support the conclusions. This is not given in the articles, which more covers what has been studied.

No references is given for the measure of the degree of aggrement. line 83. What action was taken with regard to lack of agreement?

The table do not report type of research as said on line 111.

On line 129-130: It is difficult to follow the text. Figure 2 is unexplaind. Please amend the legend to the figure. What is on the vertical axes and what is the message? That the majorities of studies came from other sources tha WOS and SCOPUS??

Minor comments:

line 40: delete the , before the brackets

line 98: what is "transversal nature"? Seldom used concept, please amend.

Table 1: the page break leads to difficulties in reading the table. Consider revision.

Line 112 and 113 is missing in the manuscript as downloaded by me.

line 123: what is the evidence for the expression: "emtional influence has been the most studied, becasue it exerts a greater influence on the spectators"?

line 148-150: The sentence is difficult to understand. Reformulate and make several sentences.

Discussion: This seems more like reporting of the results of the different studies and not a discussion of the presented results in the paper. 

Author Response

Dear the editor and reviewers,

We would like to express our gratitude for the time taken to review this manuscript and for the comments made, which we believe to be critical for producing rigorous and quality research. We have detailed below the changes made in the original article: ijerph-399330

Modifications have been made in the original manuscript following the reviewers’ comments. For each modification we have written: the original comment as written by the reviewer in addition to the page and line number; and the change made in response to that comment. Changes have been made using the tool “Track changes” enabling editor and reviewers to identify modifications easily.

MODIFICATIONS

EDITOR

Dear editor, first thank you for your suggestions for improving the article.

In addition, our manuscript has been reviewed by a native scientist.

To make the changes, all the authors have read the text conscientiously and made the changes the reviewers have said.

We remain attentive to your response.

REVIEWER 1

Comment 1:

The paper is a scoping review. No information is given on the strenghts of the conclusions. Some studies were rather small. No meta-analysis was done probably because the diversity of designs, data collected and samples

Response 1:

Dear reviewer, first of all, we would like to thank you for the time you have devoted to reviewing this manuscript and the suggestions you have made to us, as we consider all of them to be crucial to improving the study.

In order for the study to be a systematic review, we have made the relevant changes by extracting the most significant data from each study. A meta-analysis was not carried out because we first want to know how the media influence the behaviour of society in general. To later carry out another meta-analysis study. Also, due to the disparity of the data, the studies we have selected were complex to perform a meta-analysis.

Comment 2:

The aims, on line 75-78, are very ambitious. The formulation impliues that quantitative/statistical analysis should support the conclusions. This is not given in the articles, which more covers what has been studied.

Response 2:

Dear reviewer, as you told us the study objective was too ambitious.

We have therefore followed your advice and modified it so that it is really in line with the analysis, results and conclusions proposed in this study (line 78).

Comment 3:

No references is given for the measure of the degree of aggrement. line 83. What action was taken with regard to lack of agreement?

Response 3:

Dear Editor, in the absence of consensus on a study, it was eliminated. Only taking those in which there was a 100% coincidence (line 83).

Comment 4:

The table do not report type of research as said on line 111

Response 4:

Dear reviewer, as you indicated we made a mistake and did not include the type of investigation.

We have modified and corrected this section to include the type of investigation.

Comment 5:

On line 129-130: It is difficult to follow the text. Figure 2 is unexplaind. Please amend the legend to the figure. What is on the vertical axes and what is the message? That the majorities of studies came from other sources tha WOS and SCOPUS??

Response 5:

Dear reviewer, thank you for your suggestions as this has clarified and improved the study.

The text has been modified to make it understandable (line 137-140).

Information has been added to make the figure clear and understandable. It has also been specified what each axis of figure 2 symbolizes.

In addition, we have clarified that all studies have been selected from WOS and SCOPUS.

Comment 6:
line 40: delete the , before the brackets

Response 6:

Dear Editor, we have made the correction that you have suggested, since it is true that in line 40 there was an error, which has already been corrected.

Comment 7:

line 98: what is "transversal nature"? Seldom used concept, please amend.

Response 7:

Dear reviewer thank you for your suggestion, as it is true that the concept used was not appropriate. Therefore we have changed transverse nature for cross-sectional which is the ideal (line 100).

Comment 8:

Table 1: the page break leads to difficulties in reading the table. Consider revision.

Response 8:

Dear reviewer, as suggested to us, we have reworked Table 1 so that it is clear and easy to understand. To this end, we have included some sections, as a type of research, and we have eliminated others that did not agree with the objective of the study as indicated to us.

Once again, I would like to thank you for your suggestions, since they are valuable for formalizing the study and making it scientifically rigorous.

Comment 9:

Line 112 and 113 is missing in the manuscript as downloaded by me.

Response 9:

Dear reviewer, we apologize if any section could not be viewed. However, we have already made the necessary changes so that this does not happen.

Comment 10:

line 123: what is the evidence for the expression: "emtional influence has been the most studied, becasue it exerts a greater influence on the spectators"?

Response 10:

Dear reviewer, thank you for the contribution which we believe has improved the manuscript.

With that phrase we meant that most of the studies we have selected as the base body refer to emotional influence. For this reason it has been modified in the manuscript so that there is no confusión (line 125).

Comment 11:

line 148-150: The sentence is difficult to understand. Reformulate and make several sentences.

Response 11:

Dear reviewer, we have corrected the lines that you have indicated to us, also we have divided it in more phrases so that the message that is transmitted is clear (line 154-158).

Comment 12:

Discussion: This seems more like reporting of the results of the different studies and not a discussion of the presented results in the paper. 

Response 12:

Dear reviewer, thank you again for your suggestions.

We have included the pertinent modifications in the discussion so that it is correct and have scientific rigor.

Dear reviewer, thank you for all the suggestions as we believe these have been necessary to achieve greater rigor in the study. We also believe that this is of interest to readers, as it covers and shows the influences that the media exert on society. These media are immersed in our daily lives and shape our behaviours, attitudes and lifestyles. This being necessary to bear in mind especially for the youngest. For what we consider to be an innovative and interesting study.

Reviewer 2 Report

This systematic review has identified a number of relevant studies of the effect of sports media on audiences. However the paper could be much improved by English editing, and particularly to clarify concepts and terminology used in the paper. It would also be helpful if the Results and Discussion were more systematically organised around the main themes of the reviewed studies. Various statements also need to be checked for accuracy (eg line 186 says that 'the media do not exert a significant influence on the acquisition of products", whereas in Table 1 it is stated that Thompson et al 2016 and Voorveld et al 2015 found increased 'desire for objects/consumption of visualised products' respectively). 

Author Response

REVIEW 2

Comment 1:

However, the paper could be much improved by English editing, and particularly to clarify concepts and terminology used in the paper.

Responds 1:

Dear reviewer, first of all, thank you for taking the time to review the manuscript so that it may be improved.

We have followed your suggestions and all the authors have reviewed the English to make it better. In addition, some terms have been modified for others that are more suitable.

Thanks again for your suggestions as we think they are enriching to improve the manuscript.

Comment 2:

It would also be helpful if the Results and Discussion were more systematically organised around the main themes of the reviewed studies.

Response 2:

Dear reviewer, as you are suggesting, we have made changes to both the results and the discussion so that the discussion focuses more specifically on the objective of the study.

Comment 3:

Various statements also need to be checked for accuracy (eg line 186 says that 'the media do not exert a significant influence on the acquisition of products", whereas in Table 1 it is stated that Thompson et al 2016 and Voorveld et al 2015 found increased 'desire for objects/consumption of visualised products' respectively).

Response 3:

It is true that we had made a mistake in that part of the text, which has already been fixed (line 204).

Dear reviewer, we have made all the changes you have suggested, as we believe they have been very enriching for the study. Likewise, we believe that it is a study that is interesting for the readers, as it shows a situation in which we are all immersed in society, which acts in us both positively and negatively.

Round 2

Reviewer 1 Report

The manuscript is now much improved.

Only two comments

It would be possible to mention in the manuscript that this a scoping review looking into what has been researched. Or in the title

It is not necessary to have the explanations both in the legend and in the text (figure 2). Keep it in the legend to the figure.

Author Response

We would like to express our gratitude for the time spent reviewing this manuscript and for the comments made, which we consider critical to producing rigorous and quality research. The changes to the original article with ID: 399330 are detailed below.

Modifications have been made to the original manuscript following the reviewers' comments. For each modification we have written: the original comment written by the reviewer and the change made in response to that comment. The changes have been made using the "Change Control" tool that allows the editor and reviewers to easily identify modifications.

 MODIFICATIONS

EDITOR

Dear Editor, first of all, thank you for the suggestions you have given us to improve the manuscript.

Also, to indicate that, in order to make the changes, all the authors have read the text conscientiously and have made the changes suggested by both reviewers.

We remain attentive to your response and to any suggestions you may wish to make.

REVIEW 1

Comment 1:

It would be possible to mention in the manuscript that this a scoping review looking into what has been researched. Or in the title

Response 1:

Dear reviewer, first of all, thank you for the suggestions you have made, because we believe that with them the manuscript will improve considerably.

As you told us, we have included in line 78 that this is a scope review.

Comment 2:

It is not necessary to have the explanations both in the legend and in the text (figure 2). Keep it in the legend to the figure

Response 2:

Dear reviewer, as you have suggested to us, we have left only the explanation of the figure in the legend of the same, removing it from the text.

Reviewer 2 Report

In my opinion the paper has been somewhat improved but would benefit further by extensive English editing and clarification of the meaning of various terms used. For example line 18 states “.. this  research has carried out a systematic review of the scientific production of international impact".This is unclear and could be simplified by stating “this  research has carried out a systematic review of international studies “; also in line 27, “ The media conditions the vision and understanding of society”, what is meant by “vision and understanding”?   Values and knowledge? Also, is ‘condition’ meant in the sense of ‘conditioning’ in a learning theory context, or just ‘influences’? line 68: what is meant by “sentimental way”? There are numerous other examples throughout the text (including the first sentence of the Conclusions and lines 290-294).

Re Table 2, these five types of ‘influence’ need to be explicitly defined (preferably with examples) in the Results section, and, as stated previously clearly identified in the Discussion. Also, might it be more accurate to call these media 'effects’ rather than media 'influences’?

Author Response

REVIEW 2

Comment 1:

In my opinion the paper has been somewhat improved but would benefit further by extensive English editing

Response 1:

Dear reviewer, first of all, thank you for your time for the study to improve. Also, tell him that we have done a style review of the study to make it correct

Comment 2:

Clarification of the meaning of various terms used. For example line 18 states “.. this  research has carried out a systematic review of the scientific production of international impact".This is unclear and could be simplified by stating “this  research has carried out a systematic review of international studies

Response 2:

Dear reviewer, as indicated by the change you suggested to us in line 18, in order to make the text clearer.

Comment 3:

Also in line 27, “ The media conditions the vision and understanding of society”, what is meant by “vision and understanding”?   Values and knowledge? Also, is ‘condition’ meant in the sense of ‘conditioning’ in a learning theory context, or just ‘influences’

Response 3:

Dear reviewer, we have modified line 27 to make it clearer. What we are referring to in that sentence is that the media are the ones that condition our way of seeing and understanding sport.

Comment 4:

Line 68: what is meant by “sentimental way”? There are numerous other examples throughout the text (including the first sentence of the Conclusions and lines 290-294).

Response 4:

Dear Editor, we have seen the error you were telling us about and therefore the full text has been revised to correct this error, as you can see in line 67.

Comment 5:

Re Table 2, these five types of ‘influence’ need to be explicitly defined (preferably with examples) in the Results section, and, as stated previously clearly identified in the Discussion. Also, might it be more accurate to call these media 'effects’ rather than media 'influences’?

Response 5:

Dear reviewer, thank you first of all for your advice, as we believe it has been very useful in clarifying some concepts.

We have also included the definition of the different types of influences (line 122).

Dear Editor, we have clarified and differentiated the different types of influence in the discussion:

- Emotional and violence influence, line 153

- Influence on health, line 169

- Influences on  physical, 183

- Influences on consumption, line 198

We believe that in this way the discussion has become much clearer and more orderly. However, if you think it is necessary for us to include sub-sections to make this paragraph clearer, please let us know.

However, we believe that we should call these means influences rather than effects. For it is these kinds of influences that subsequently generate various effects in society.

Dear reviewer, we believe that the publication of this study may be of interest to readers, as it is a topical issue that increasingly influences society and especially the youngest. It also shows the results of scientific research over the last decade. Through which it can be observed how the sports media can benefit or harm society.

Round 3

Reviewer 2 Report

In my opinion the paper needs comprehensive English translation, particularly to ensure that the terminology has the correct meaning.